# Data-driven drug-induced QT prolongation surveillance using adverse reaction signals derived from 12-lead and continuous electrocardiogram data

**Byung Jin Choi[1], Yeryung Koo[2], Tae Young Kim[2], Hong-Seok Lim[3], Dukyong Yoon[2,4,5]***

1 Department of Biomedical Informatics, Ajou University School of Medicine, Suwon, Gyeonggi-do, Republic of Korea, 2 BUD.on Inc, Jeonju, Jeollabuk-do, Republic of Korea, 3 Department of Cardiology, Ajou University School of Medicine, Suwon, Gyeonggi-do, Republic of Korea, 4 Department of Biomedical Systems Informatics, Yonsei University College of Medicine, Yongin, Gyeonggi-do, Republic of Korea, 5 Center for Digital Health, Yongin Severance Hospital, Yonsei University Health System, Yongin, Gyeonggi-do, Republic of Korea

These authors contributed equally to this work.
* dukyong.yoon@yonsei.ac.kr

**Data Availability Statement:** Data for drugs known to prolong QT interval can be downloaded at CredibleMeds (https://www.crediblemeds.org/). Data for analysis on the 12-lead ECGs is based on

## Abstract

Drug-induced QT prolongation is one of the most common side effects of drug use and can cause fatal outcomes such as sudden cardiac arrest. This study adopts the data-driven approach to assess the QT prolongation risk of all the frequently used drugs in a tertiary teaching hospital using both standard 12-lead ECGs and intensive care unit (ICU) continuous ECGs. We used the standard 12-lead ECG results (n = 1,040,752) measured in the hospital during 1994–2019 and the continuous ECG results (n = 4,835) extracted from the ICU's patient-monitoring devices during 2016–2019. Based on the drug prescription frequency, 167 drugs were analyzed using 12-lead ECG data under the case-control study design and 60 using continuous ECG data under the retrospective cohort study design. Whereas the case-control study yielded the odds ratio, the cohort study generated the hazard ratio for each candidate drug. Further, we observed the possibility of inducing QT prolongation in 38 drugs in the 12-lead ECG analysis and 7 drugs in the continuous ECG analysis. The seven drugs (vasopressin, vecuronium, midazolam, levetiracetam, ipratropium bromide, nifedipine, and chlorpheniramine) that showed a significantly higher risk of QT prolongation in the continuous ECG analysis were also identified in the 12-lead ECG data analysis. The use of two different ECG sources enabled us to confidently assess drug-induced QT prolongation risk in clinical practice. In this study, seven drugs showed QT prolongation risk in both study designs.

the publicly available ECG-ViEW II Database (Kim et al. "ECG-ViEW II, a freely accessible electrocardiogram database." PLoS One 2017 Apr 24;12(4):e0176222). Fully anonymized data can be obtained by submitting an adequate research plan and CITI certification (http://www.ecgview.org). Data for analysis on the continuous ECG is based on highly sensitive intensive care unit patients' data and cannot be opened to the public without approval from The Institutional Review Board of Ajou University Hospital (ajou_irb@aumc.ac.kr). The minimal data for supporting the results in this manuscript is provided in Table 1 and supplementary tables in Supporting information.

**Funding:** This work was supported by the Korea Medical Device Development Fund grant funded by the Korea government (the Ministry of Science and ICT; Ministry of Trade, Industry, and Energy; Ministry of Health & Welfare; and Ministry of Food and Drug Safety) and awarded to DY (Project Number: 1711138152, KMDF_PR_20200901_0095). This research was also supported by a grant from Ministry of Food and Drug Safety in 2020, awarded to DY (19182MFDS406). BUD.on Inc. provided support in the form of salaries for authors YK and TYK, but did not have any additional role in the study design, data collection and analysis, decision to publish, or preparation of the manuscript. The specific roles of these authors are articulated in the 'author contributions' section. BUD.on Inc. had no role in study design, data collection and analysis, decision to publish, or preparation of the manuscript.

**Competing interests:** BJC, and HSL declare that they have no competing interests. DY is the founder of BUD.on Inc. and YK and TYK is an employee of BUD.on Inc. BUD.on Inc. did not have any role in the study design, analysis, decision to publish, or the preparation of the manuscript. This does not alter our adherence to PLOS ONE policies on sharing data and materials.

## Introduction

The prolongation of the QT interval refers to the extension of the interval between the start of the QRS complex and the end of the T wave by external factors. The delay in ventricular repolarization caused by a reduction in the outward potassium current results in the broadening of ventricular action potentials and, consequently, the prolongation of the QT interval [1]. QT prolongation may cause diverse arrhythmic conditions, including torsade de pointes, which is a type of ventricular tachycardia known to cause sudden cardiac death [2–4].

Drug-induced QT prolongation is the most common cause of acquired QT prolongation [5,6]. Furthermore, A drug's propensity to cause QT interval prolongation may cause its withdrawal from the market [7]. Accordingly, the early detection of drug-induced QT prolongation is crucial from the medical and socioeconomic perspectives.

Many studies have examined drug-induced QT prolongation, but most of these investigations have the following limitations. In many cases, researchers first select the drug to be investigated [5,8]. This approach enables researchers to investigate the risks for only some drugs of interest. Hence, such risk assessment studies often exclude drugs that are known to be less lethal, such as those used in chronic disease or symptomatic treatments. As the risk of QT prolongation is higher in patients with chronic disease than in the general population [9,10], comprehensive studies on the QT prolongation risks of the relevant less lethal drugs are required. To reduce this selection bias, we selected hundreds of candidate drugs, including those that had not been studied before, using a data-driven approach according to the drugs' frequency of use in a tertiary teaching hospital.

The second limitation of existing studies is that they are based on a standard 12-lead ECG alone. The measuring time for standard 12-lead ECGs is only 10 seconds at a time, and there are inconsistent days to years of time gaps between the two measurements. Since a patient's drug adherence for the period between two measurements is unknown, it is difficult to capture the exact time gap between drug exposure and adverse drug events and identify acute adverse drug events. This causes researchers to hesitate in performing retrospective cohort studies with standard 12-lead ECG data.

Therefore, in addition to using standard 12-lead ECG data, this study used the continuous ECG data extracted from an intensive care unit (ICU) patient-monitoring device. As the administration of all drugs was recorded in the ICU, we accurately identified the exact time point of drug exposure and the time until the QT prolongation event. Further, using continuous ECG data, we conducted a retrospective cohort study, which is considered to obtain higher levels of evidence than case-control studies, and investigated acute adverse drug events. However, it has limited generalizability because the continuous ECG was usually measured in an intensive care unit (ICU), and the characteristics of patients and popularly used drugs are different between ICU and general wards.

To make up for the limitation of standard 12-lead ECG data and the continuous ECG data extracted from an ICU patient-monitoring device, we used both data together in evaluating the QT prolongation risk of drugs. We built two separate algorithms to analyze the risk of QT interval prolongation by adopting the data-driven approach and conducted a case-control study using standard 12-lead ECG data and a retrospective cohort study using continuous ECG data.

## Materials and methods

The Institutional Review Board of Ajou University Hospital approved this study (IRB No. AJIRB-MED-MDB-19-406) and waived the requirement for informed consent because the study retrospectively used anonymized data.

## Dataset

We used two ECG sources. The first comprised standard 12-lead ECG data [11]. We extracted the standard 12-lead ECG data from the ECG MUSE system (GE Healthcare) of Ajou University Hospital for the period between 1994 and 2018 and linked them to electronic health record (EHR) data. Accordingly, we linked the 1,040,752 12-lead ECG data to the diagnosis, prescription, and procedure records of 447,632 patients. Since we extracted data regardless of visit or admission type, all patients (outpatients, inpatients admitted to general wards/ICUs, patients visited emergence department, etc.) were included.

The second ECG source included continuous ECG data. We extracted continuously monitored ECG data from the patient-monitoring devices (Philips [Amsterdam, The Netherlands] and Nihon Kohden [Tokyo, Japan]) of 4,835 patients in the hospital's ICU for the period between 2016 and 2018. Further, we calculated the QT interval from the raw signal using the method proposed by Chesnokov algorithm et al. [12], corrected it using the Bazett formula [13] to obtain the QTc value, and recorded the median QTc interval for every 10 seconds. Finally, we linked the data to EHR for analysis.

As a reference for drugs known to prolong QT interval, we used the QT risk drug list provided by CredibleMeds [14], since this list is commonly used as a reliable reference in QT prolongation studies [15,16]. The list classifies drugs into four categories: the drugs that should be avoided in treating congenital long QT syndrome, those with a known risk of QT prolongation, those with a possible risk of QT prolongation, and those with a conditional risk of QT prolongation.

## Case-control analysis using 12-lead ECG data

**Candidate drug selection.** To create the list of candidate drugs for analysis, we extracted all ECG results with QT prolongation (QTc>450 for men and QTc>460 for women) and randomly selected one ECG with several prolonged QT intervals for a single patient [17–19]. Further, we extracted all prescription data seven days before the date of ECG measurements in the QT prolongation cases, counting the frequency of each drug use. We also included all drugs to evaluate the QT prolongation risk of all drugs, including those not unknown yet. However, drugs prescribed more than 500 times in the cases were included to secure the statistical power. To reduce indication bias, we excluded the prescriptions ordered on the ECG measurement dates. Finally, based on the QT risk drug list provided by CredibleMeds [14], the candidate drugs were divided into subgroups with four levels by a clinician: Rank 4, 3, 2, and 1 drugs have a known risk, a possible risk, a conditional risk, and an unknown risk of QT prolongation, respectively.

**Study design and population.** As shown in Fig 1, we randomly selected one ECG result per patient (n = 447,632). After excluding the ECG results without gender or age information or with age outliers (n = 3,050), we included 444,582 ECG results in the study. Further, we adopted the propensity score matching method to match the control group (subjects whose QTc interval was within normal range) with the case group (subjects whose QTc interval was prolonged) to adjust confounding variables with the following covariates: Gender and age at the ECG examination date, the latest serum potassium and calcium levels calculated within a year of the ECG measurement date, the comorbidities recorded in the EHR within a year of the ECG measurement date (e.g., myocardial infarction, congestive heart failure, ischemic stroke, hemorrhagic stroke, diabetes mellitus, hypothyroidism, renal disease, AIDS/HIV, alcohol abuse, drug abuse, liver disease, and severe liver disease), and the frequency of drug use for each drug rank group within seven days of the ECG date. S1 Table provides the complete drug list of each drug rank that was used in counting the frequency. For patients without laboratory

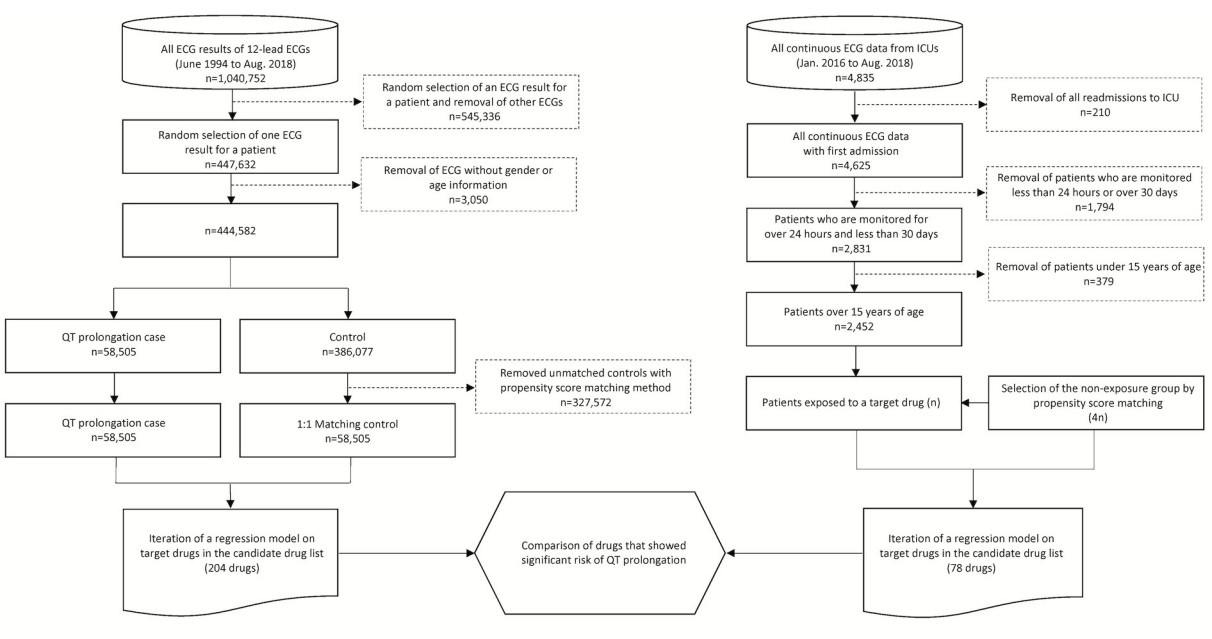

**Fig 1. Flow chart of the research process.**

test results dated within one year of the ECG examination date, we replaced the missing values with the median values of patients of the same age group divided by 10-year intervals.

## Retrospective cohort analysis using continuous ECG data

**Study design and population.** We performed a survival analysis based on a retrospective cohort study using continuous ECG data to identify drugs having QT prolongation risk. We excluded all patients with records indicating ICU hospitalization prior to the study period (n = 311) from the analysis. Similarly, we excluded all patients who had ECG monitoring data with durations less than 24 hours or more than 30 days (n = 1,794). We excluded all patients under 15 years of age (n = 379), as well.

To secure at least 5% of the exposure group for further analysis, we selected 78 drugs prescribed for more than 5% of the study subjects as candidate drugs. Eighteen drugs were already classified as QT prolongation risk drugs by CredibleMeds; these are used for propensity score matching (see "Matching exposure and non-exposure groups"). Therefore, we finally analyzed the remaining 60 drugs.

**Definition of index time and other variables.** We defined the first drug initiation time of a target drug for each patient as the drug's index time. We observed the QT intervals for 12 hours from the index time to assess the QT prolongation risk. For each covariate, all the drug infusions within 12 hours of the index time and all medical diagnosis records made before the index time were extracted from EHR data. The laboratory results of calcium and potassium levels before and after seven days of the index time were also extracted.

We defined QT prolongation as QTc>450 in men and QTc>460 in women [17–19]. Further, we excluded all patients who showed QT prolongation within 5 minutes before the index time. Finally, we measured the duration from the index time to the time of occurrence of QT prolongation for the survival analysis.

**Matching exposure and non-exposure groups.** We matched the drug-exposed and unexposed groups (or exposure and non-exposure groups) to eliminate the influence of confounding variables. Since a patient's condition may vary during his or her hospital stay, we first sliced the continuous ECG data of the non-exposure group into 12-hour-long segments and selected the segment having the closest length-of-stay to the length-of-stay at the index time of the exposure group.

To match the groups, we applied the propensity score matching method with the following covariates: gender, age, ward type, calcium and potassium levels (closest laboratory record with the index time), 18 for known QT prolongation risk [14] and used in more than 5% of ICU patients, and 9 QT-related clinical factors of more than 1% of ICU patients. Each patient in the exposure group was matched to four patients in the non-exposure group. S2 Table shows the complete list of drugs and comorbidities used in propensity score matching.

## Statistical analysis

We first compared subjects' demographic characteristics (gender and age), laboratory test results (calcium and potassium levels), and comorbidities among the two study designs using Pearson's chi-square test (for categorical data) and the independent two-sample t-test (for continuous data). Further, for the case-control analysis based on 12-lead ECG data, we performed conditional logistic regressions for each drug in the candidate drug list. Subsequently, we estimated the odds ratios (ORs) and 95% confidence intervals (CIs) of each drug with a significance level of $p < 0.05$. For the retrospective cohort analysis based on continuous ECG data, we established the Cox Proportional-Hazards Model to calculate the hazard ratios (HRs) of QT prolongation for each drug and 95% CIs of each drug at a significance level of $p < 0.05$.

Further, to validate our methods, we analyzed the QT prolongation risk of each drug in the drug list provided by CredibleMeds as the positive control. While analyzing the drugs in the QT drug list, we excluded the drug from the matching variable in propensity score matching and calculated the variable's ORs or HRs.

To correct the multiple comparison problem, we calculate the false discovery rate and validated the results at a significance level of $p < 0.05$. Data management was performed using Azure data studio version 1.19.0, and all statistical analyses were conducted using R version 4.0.2.

## Results

### Baseline characteristics

Table 1 depicts the study subjects' baseline characteristics prior to propensity score matching for the case-control analysis and the subjects' characteristics for the retrospective cohort analysis. In the case-control analysis, the mean age of subjects was higher in the case group than in the control group, and the laboratory test results (calcium and potassium levels) were higher in the control group than in the case group. Further, comorbidities, except hypothyroidism and AIDS, were higher in the case than in the control group. After propensity score matching, 58,505 QT prolongation cases and their matched 58,505 controls were enrolled in the analysis for the case-control analysis.

In the retrospective cohort analysis, we finally selected 2,351 patients for the analysis, and according to the target, those patients were subdivided into the non-exposure group and the exposure group. Table 1 shows the baseline characteristics of subgroups according to whether QT prolongation was observed at least once or not. The age, incidences of diabetes, sudden cardiac arrest, sepsis, and congestive heart failure were higher in the patients who ever have experienced QT prolongation at least one time.

**Table 1. Baseline characteristics of subjects.**

| | QT prolongation | | | | | |
| --- | --- | --- | --- | --- | --- | --- |
| | Case-control analysis | | | Retrospective cohort analysis | | |
| | Control group | Case group | p-value | No. of QT prolongation $<$ 1 | No. of QT prolongation $\geq$ 1 | p-value |
| N | 386,077 | 58,505 | | 1,371 | 980 | |
| Gender, male, n (%) | 189,146 (49.0) | 29,856 (51.0) | <0.001 | 997 (72.7) | 635 (64.8) | <0.001 |
| Age, mean (SD) | 42.4 (20.4) | 55.1 (20.8) | <0.001 | 52.3 (20.6) | 59.7 (18.8) | <0.001 |
| Potassium, mean (SD) | 4.1 (0.4) | 4.0 (0.6) | <0.001 | 3.9 (0.8) | 3.9 (1.0) | 0.251 |
| Calcium, mean (SD) | 9.2 (0.6) | 8.8 (0.8) | <0.001 | 8.0 (1.6) | 7.9 (1.7) | 0.086 |
| Myocardial infarction, n (%) | 3,457 (0.9) | 1,723 (2.9) | <0.001 | 60 (4.4) | 54 (5.5) | 0.244 |
| Congestive heart failure, n (%) | 2,390 (0.6) | 1,812 (3.1) | <0.001 | 15 (1.1) | 33 (3.4) | <0.001 |
| Ischemic stroke, n (%) | 6,332 (1.6) | 2,519 (4.3) | <0.001 | 41 (3.0) | 73 (7.4) | <0.001 |
| Hemorrhagic stroke, n (%) | 2,207 (0.6) | 1,480 (2.5) | <0.001 | 279 (20.4) | 142 (14.5) | <0.001 |
| Diabetes mellitus, n (%) | 12,364 (3.2) | 3,284 (5.6) | <0.001 | 40 (2.9) | 53 (5.4) | 0.003 |
| Renal disease, n (%) | 3,074 (0.8) | 2,068 (3.5) | <0.001 | 31 (2.3) | 41 (4.2) | 0.011 |
| Hypothyroidism, n (%) | 1,999 (0.5) | 255 (0.4) | 0.010 | NA | NA | NA |
| AIDS/HIV, n (%) | 123 (0.0) | 22 (0.0) | 0.552 | NA | NA | NA |
| Alcohol abuse, n (%) | 1,145 (0.3) | 921 (1.6) | <0.001 | NA | NA | NA |
| Drug abuse, n (%) | 392 (0.1) | 199 (0.3) | <0.001 | NA | NA | NA |
| Liver disease, n (%) | 1,717 (0.4) | 1,350 (2.3) | <0.001 | NA | NA | NA |
| Severe liver disease, n (%) | 282 (0.1) | 424 (0.7) | <0.001 | NA | NA | NA |
| Sepsis, n (%) | NA | NA | NA | 16 (1.2) | 23 (2.3) | 0.041 |
| Sudden cardiac arrest, n (%) | NA | NA | NA | 16 (1.2) | 51 (5.2) | <0.001 |
| AV block, n (%) | NA | NA | NA | 19 (1.4) | 17 (1.7) | 0.611 |

AV block, atrioventricular block; NA, not applicable; SD, standard deviation.

## Statistical analysis results

**Case-control analysis.** The drug selection process identified the following candidate drugs: 167 rank 1, 15 rank 2, 8 rank 3, and 14 rank 4 drugs. 64.29% of the rank 4 drugs showed a QT prolongation risk with a significance level $p < 0.05$; this is the highest percentage among all ranking groups. As shown in Fig 2, the percentages of drugs with QT

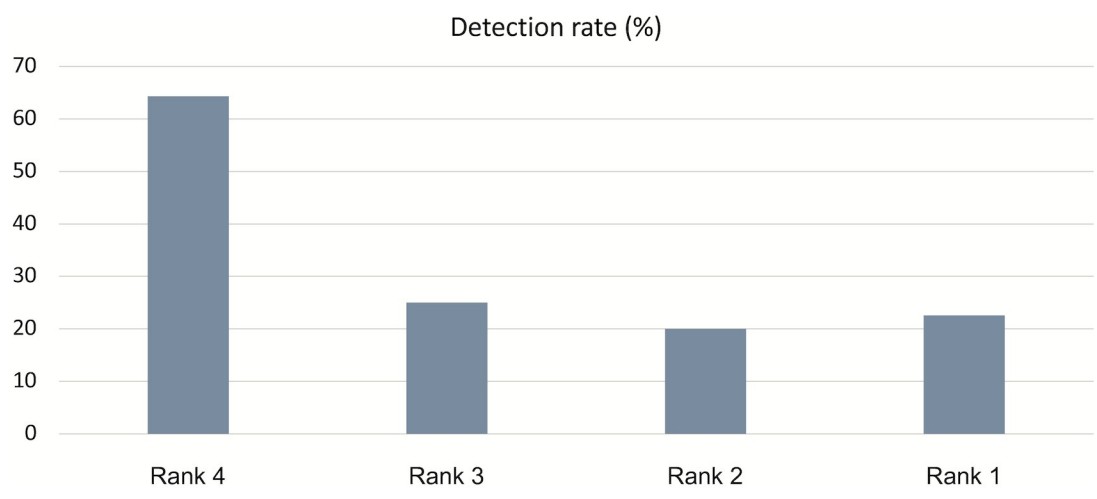

**Fig 2. Results of the positive rates of each drug group calculated to validate the algorithm.**

prolongation risks in each ranking group were consistent with the order of QT risk levels indicated by CredibleMeds.

As shown in Table 2, 38 rank 1 drugs (i.e., drugs with unknown risk of QT prolongation) showed significant QT prolongation risk at the $p < 0.05$ significance level. The five drugs with the highest risks of QT prolongation were an antidiuretic hormone (OR [95% CI], 2.05 [1.97–2.13]); somatostatin, a growth hormone–inhibiting hormone (OR [95% CI], 1.94 [1.77–2.11]);

**Table 2. Results of 38 rank 1 drugs with significant QT prolongation risks.**

| Drug | OR | CI (95%) | *p*-value |
|---|---|---|---|
| Vasopressin | 2.05 | 1.97–2.13 | <0.001 |
| Somatostatin | 1.94 | 1.77–2.11 | <0.001 |
| Etomidate | 1.81 | 1.77–1.85 | <0.001 |
| Methylergometrine | 1.80 | 1.68–1.92 | <0.001 |
| Lorazepam | 1.79 | 1.74–1.84 | <0.001 |
| Vecuronium Bromide | 1.68 | 1.63–1.73 | <0.001 |
| Hydrocortisone | 1.67 | 1.56–1.78 | <0.001 |
| Ceftriaxone | 1.65 | 1.61–1.69 | <0.001 |
| Ipratropium Bromide | 1.64 | 1.6–1.68 | <0.001 |
| Levetiracetam | 1.64 | 1.58–1.7 | <0.001 |
| Perindopril | 1.55 | 1.48–1.62 | <0.001 |
| Labetalol | 1.55 | 1.49–1.61 | <0.001 |
| Ceftazidime | 1.52 | 1.42–1.62 | <0.001 |
| Rosuvastatin | 1.35 | 1.31–1.39 | <0.001 |
| Carvedilol | 1.34 | 1.29–1.39 | <0.001 |
| Morphine | 1.32 | 1.26–1.38 | <0.001 |
| Spironolactone | 1.31 | 1.25–1.37 | <0.001 |
| Chlorpheniramine | 1.31 | 1.29–1.33 | <0.001 |
| Isosorbide Dinitrate | 1.30 | 1.27–1.33 | <0.001 |
| Clopidogrel | 1.25 | 1.22–1.28 | <0.001 |
| Remifentanil | 1.18 | 1.13–1.23 | <0.001 |
| Midazolam | 1.18 | 1.09–1.27 | <0.001 |
| Propacetamol | 1.17 | 1.09–1.23 | <0.001 |
| Ibuprofen | 1.24 | 1.17–1.31 | 0.0028 |
| Ramipril | 1.17 | 1.11–1.23 | 0.0028 |
| Hydralazine | 1.49 | 1.36–1.62 | 0.003 |
| Captopril | 1.31 | 1.21–1.41 | 0.0062 |
| Levocloperastine | 1.25 | 1.17–1.33 | 0.0065 |
| Ticagrelor | 1.32 | 1.22–1.42 | 0.0093 |
| Clindamycin | 1.16 | 1.05–1.27 | 0.0095 |
| Theobromine | 1.23 | 1.15–1.31 | 0.0124 |
| Nifedipine | 1.17 | 1.03–1.31 | 0.0133 |
| Cefotetan | 1.11 | 1.03–1.19 | 0.0233 |
| Valproate | 1.14 | 1.01–1.27 | 0.0453 |
| Tiotropium | 1.31 | 1.17–1.45 | 0.0455 |
| Propranolol | 1.18 | 1.05–1.31 | 0.0458 |
| Erdosteine | 1.25 | 1.13–1.37 | 0.0486 |
| Cefpiramide | 1.12 | 1.05–1.19 | 0.0487 |

CI, confidence interval; OR, odds ratio.

**Table 3. Hazard and odds ratios for seven drugs with significant QT prolongation risk.**

| | Retrospective study | | | Case-control study | | |
|---|---|---|---|---|---|---|
| | **HR** | **CI (95%)** | ***p*-value** | **OR** | **CI (95%)** | ***p*-value** |
| Vasopressin | 1.49 | 1.33–1.65 | 0.024 | 2.05 | 1.97–2.13 | <0.001 |
| Vecuronium | 1.76 | 1.53–1.99 | 0.021 | 1.68 | 1.63–1.73 | <0.001 |
| Midazolam | 1.37 | 1.27–1.47 | 0.028 | 1.18 | 1.16–1.21 | <0.001 |
| Levetiracetam | 1.43 | 1.25–1.61 | <0.001 | 1.51 | 1.3–1.72 | <0.001 |
| Ipratropium bromide | 1.4 | 1.32–1.48 | <0.001 | 1.64 | 1.60–1.68 | <0.001 |
| Nifedipine | 1.33 | 1.16–1.5 | 0.008 | 1.17 | 1.03–1.31 | 0.005 |
| Chlorpheniramine | 1.06 | 1.02–1.1 | 0.008 | 1.31 | 1.29–1.33 | <0.001 |

HR, hazard ratio; OR, odds ratio.

etomidate, a short-acting intravenous anesthetic agent (OR [95% CI], 1.81 [1.77–1.85]); methylergometrine, a smooth muscle constrictor (OR [95% CI], 1.8 [1.68–1.92]); and lorazepam, a benzodiazepine acting on the brain and nerves (OR [95% CI], 1.79 [1.74–1.84]). S3 Table depicts the complete results for all ranking groups.

**Retrospective cohort analysis.** Vasopressin (HR [95% CI], 1.49 [1.33–1.65]), vecuronium (1.76 [1.53–1.99]), midazolam (1.76 [1.53–1.46]), levetiracetam (1.43 [1.25–1.61]), ipratropium bromide (1.4 [1.32–1.48]), nifedipine (1.33 [1.16–1.5]), and chlorpheniramine (1.06 [1.02–1.1]) showed significant QT prolongation risks at p < 0.05 (Table 3). These seven drugs revealed significant QT prolongation risks in the case-control study, as well. Among 18 drugs on the drug list provided by CredibleMeds, 12 (66%) showed significant QT prolongation risk. S4 Table depicts the complete analysis results.

## Discussion

This study adopted a data-driven approach and used two different ECG sources to analyze the QT prolongation potential of 167 drugs using standard 12-lead ECG data and 60 drugs using continuous ECG data. It revealed the possibility of inducing QT prolongation in 38 drugs in the standard 12-lead ECG analysis and 7 in the ICU continuous ECG analysis.

QT prolongation is one of the most well-known side effects of drug use [5,6,20,21], and numerous studies have been conducted on this aspect [22]. Nevertheless, studies on the possibility of prolonging QT side effects in clinical practice remain insufficient because such studies generally focus on only a limited number of drugs selected by clinicians. Since clinicians are primarily interested in only a few drugs associated with the diseases treated by them, they may not consider drugs without specific indications. To reduce this bias, we selected candidate drugs based only on the number of prescriptions in our study. Therefore, we could observe the possibility of inducing QT prolongation even in drugs prescribed for conservative treatment, such as vecuronium and naproxen.

Many earlier studies have the limitation that they only used 12-lead ECG data for analysis [14,23]. It is difficult to identify acute adverse drug effects in standard 12-lead ECG data analysis due to the short ECG measurement time and the large time gap between drug administration and ECG measurement. In this study, continuous ECG data were extracted from ICU patient-monitoring devices during the period from hospitalization to discharge. Hence, we could analyze ECG data both before and after drug administration. Further, by using continuous ECG data, we identified the acute adverse drug effects that could occur within 12 hours of drug initiation. An ICU patient's hospital stay is as short as 3–7 days; however, ICU patients are highly likely to suffer severe conditions and fatal complications, such as sudden cardiac

arrest following QT prolongation [24]. Therefore, it is essential to investigate the occurrence of acute adverse drug effects during patients' ICU stay.

This study has the following limitations: First, the detection of QT prolongation risk can be confounded by drug–drug interaction (DDI). In particular, DDI can be a major issue in the ICU [25]. Second, the study used a database comprising data that were retrospectively collected within a single institution. Future studies should perform multicenter and multinational research to obtain more comprehensive results. Third, the study did not account for indication bias. Indication bias refers to the case where QT prolongation occurs when drugs are prescribed to treat a specific condition that may cause QT prolongation, even though the drugs did not have any specific QT prolongation effect. Lastly, not all drugs used in the subject hospital were analyzed. It was to secure the least number of patients exposed to the target drug. If the drugs are more used, those drugs will be able to be included in our analysis in the future.

## Conclusions

In this study, we analyzed the possibilities of QT interval prolongation of drugs by adopting a data-driven approach and using two large ECG sources, standard 12-lead ECG data (n = 444,582) and continuous ECG data (n = 2452). Consequently, we observed QT interval prolongation risk in 38 out of 167 drugs with unknown risk in the candidate drug list in a case-control study based on a standard 12-lead ECG database and 7 out of 60 drugs in a retrospective cohort study based on a continuous ECG database.

## Supporting information

**S1 Table. Complete drug list for each drug rank used in counting the frequency of drug use for each drug ranking group within seven days before the ECG measurement date.** Rank 1 drugs are candidate drugs with unknown QT prolongation risk in the case-control study. The drugs in other ranks were analyzed to validate the study.
(DOCX)

**S2 Table. Complete list of the comorbidities and drugs used in the propensity score matching process in survival analysis.** The QT drug list is based on the QT risk drug list provided in CredibleMeds.org.
(DOCX)

**S3 Table. Complete results for all drugs in each of the four drug ranks.** The candidate drug list is the list of drugs in rank 1; the drugs in other ranks were analyzed to validate the study. The drugs in each rank were used to count the drug use frequency of each rank.
(DOCX)

**S4 Table. Complete analysis results of the 78 drugs considered in the survival analysis based on the continuous ECG database.**
(DOCX)

## Author Contributions

**Conceptualization:** Dukyong Yoon.

**Data curation:** Byung Jin Choi, Yeryung Koo.

**Investigation:** Byung Jin Choi, Yeryung Koo.

**Methodology:** Byung Jin Choi.

**Project administration:** Tae Young Kim.

**Supervision:** Hong-Seok Lim, Dukyong Yoon.

**Validation:** Tae Young Kim.

**Writing – original draft:** Byung Jin Choi, Yeryung Koo, Hong-Seok Lim, Dukyong Yoon.

**Writing – review & editing:** Dukyong Yoon.

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
