## [Decision Letter · Decision Letter 0]

6 Dec 2021

PONE-D-21-15865

Data-driven drug-induced QT prolongation surveillance using adverse reaction signals derived from 12-lead and continuous electrocardiogram data

PLOS ONE

Dear Dr. Yoon,

Thank you for submitting your manuscript to PLOS ONE. After careful consideration, we feel that it has merit but does not fully meet PLOS ONE’s publication criteria as it currently stands. Therefore, we invite you to submit a revised version of the manuscript that addresses the points raised during the review process.

The paper is clearly written and the methods well described. We suggest to explain why the socalled credmeds durgs were excluded.

We look forward to receiving your revised manuscript.

Kind regards,

Chiara Lazzeri

Academic Editor

PLOS ONE

Journal Requirements:

"DY was supported by the Korea Medical Device Development Fund grant funded by the Korea government (the Ministry of Science and ICT, the Ministry of Trade, Industry and Energy, the Ministry of Health & Welfare, the Ministry of Food and Drug Safety) (Project Number: 202012B04). (www.https://www.kmdf.org/)

DY was also supported by a Government-wide R&D Fund project for infectious disease research (GFID), Republic of Korea (grant numbers: HG18C0067). (https://www.gfid.or.kr)

We note that one or more of the authors are employed by a commercial company: "BUD.on Inc, Jeonju, Jeollabuk-do, Republic of Korea"

3. We note you have included a table to which you do not refer in the text of your manuscript. Please ensure that you refer to Table 3 in your text; if accepted, production will need this reference to link the reader to the Table.

Reviewers' comments:

Reviewer's Responses to Questions

**Comments to the Author**

1. Is the manuscript technically sound, and do the data support the conclusions?

Reviewer #1: Yes

2. Has the statistical analysis been performed appropriately and rigorously? 

Reviewer #1: I Don't Know

3. Have the authors made all data underlying the findings in their manuscript fully available?

Reviewer #1: No

4. Is the manuscript presented in an intelligible fashion and written in standard English?

Reviewer #1: Yes

5. Review Comments to the Author

Reviewer #1: I read this paper with interest as it aims to evaluate the high risk issue of drug-induced QT prolongation. The data driven approach described by the authors has merits in that it could overcome limitations of hypothesis-driven investigations that rely on an a priori signal. However, this approach the authors take introduces different biases, notably that infrequently prescribed drugs are not evaluated which could be strongly associated with ADEs.

Suggestions for improvement are described as follows -

1. Others relied on evidence/data from the literature to understand which are more likely to be associated with ADEs including QTc prolongation. What makes your approach unique is you did not limit yourself to those drugs with known associations. I would make this differentiation more clear in the description of candidate drug selection.

2. In the introduction there is discussion that continuous ECG monitoring is better because it allows for better understanding of causality and controls for things such as patient adherence. While this may be true, it limits the generalizability of the findings in real world situations beyond the ICU. In the real-world, outside of ICU settings patient sporadic adherence to meds is important to factor in – whether they doubled the dose or skipped a dose can significantly impact results. Please make it clear RE: what settings your findings are generalizable to and the limitations of this approach.

3. Some results are interspersed within the methods, notably in the study design and population section. Please separate the methods from the results.

4. The methods states drugs of interest are included if they are prescribed more than 500 times and then later it says for >5% of study subjects. Can you please clarify? And please elaborate on how this approach has limitations as well. Although a drug may not be frequently prescribed, its risk of prolonging the QT interval persists and you excluded drugs infrequently prescribed.

5. Please explain why the 18 drugs already classified by credmeds were excluded. Including them may have provided some triangulation to support your findings.

6. Please provide a reference for your definitions of QTc prolongation. Clinical and research audiences often look to >/=500ms as a meaningful measure, but this does not necessarily align with all measures of abnormal QTc intervals.

7. Please explain why the 12 lead data was used in addition to the continuous data.

8. In the methods, please elaborate on the patient population from which 12-lead data was collected. Was is only ICU patients as was the case for continuous data?

9. Please clarify whether the candidate drugs were evaluated before the ECG measurements (not 7 days after).

10. 1st sentence of the 3rd paragraph of the introduction: It is unclear why “Standard 12-lead electrocardiogram (ECG) data” is explicitly called out. The subsequent limitations are not related to the 12-lead (versus continuous).

11. Please be sure you are using the acronym EMR correctly. An EMR and EHR are different and most health systems use an EHR, not EMR.

12. Please consider whether adherence or compliance is the more accurate term to describe how/if patients take medications.

6. PLOS authors have the option to publish the peer review history of their article (what does this mean?). If published, this will include your full peer review and any attached files.

Reviewer #1: No

---

## [Author Response · Author response to Decision Letter 0]

3 Jan 2022

We would like to thank Reviewer 1 for the effort to review our paper and for valuable comments which have helped us substantially improve the manuscript. Our responses to the reviewer’s comments are provided below.

Responses to Reviewer 1

Comment #1: 

Others relied on evidence/data from the literature to understand which are more likely to be associated with ADEs including QTc prolongation. What makes your approach unique is you did not limit yourself to those drugs with known associations. I would make this differentiation more clear in the description of candidate drug selection.

Response to comment #1:

We would like to thank the reviewer for this suggestion. We agree with this recommendation. Following the comment, we modified the description in the “candidate drug selection” section to clarify our strength as follows:

• (Page 6) “And we included all drugs to evaluate the QT prolongation risk of all drugs, including those not unknown yet, but drugs that were prescribed more than 500 times in the cases were included to secure the statistical power.”

Comment #2 and #7: 

#2: In the introduction there is discussion that continuous ECG monitoring is better because it allows for better understanding of causality and controls for things such as patient adherence. While this may be true, it limits the generalizability of the findings in real world situations beyond the ICU. In the real-world, outside of ICU settings patient sporadic adherence to meds is important to factor in – whether they doubled the dose or skipped a dose can significantly impact results. Please make it clear RE: what settings your findings are generalizable to and the limitations of this approach.

#7: Please explain why the 12 lead data was used in addition to the continuous data.

Response to comment #2 and #7:

We appreciate the reviewer's comment. Twelve lead ECG and continuous ECG have their unique advantages and limitations. By using both, we aimed to achieve the advantages of two different data sources and overcome the limitations of each data source. We modified our description to make clear our aim of study design as follows:

• (Page 4) Therefore, in addition to using standard 12-lead ECG data, this study used the continuous ECG data extracted from an intensive care unit (ICU) patient-monitoring device. As the administration of all drugs was recorded in the ICU, we accurately identified the exact time point of drug exposure and the time until the QT prolongation event. Further, using continuous ECG data, we conducted a retrospective cohort study, considered to obtain higher levels of evidence than case-control studies, and investigated acute adverse drug events. However, it has limited generalizability because the continuous ECG was usually measured in an intensive care unit (ICU), and the characteristics of patients and popularly used drugs are different between ICU and general wards.

• (Page 4) To make up for the limitation of standard 12-lead ECG data and the continuous ECG data extracted from an ICU patient-monitoring device, we used both data together in evaluating the QT prolongation risk of drugs.

Comment #3: 

Some results are interspersed within the methods, notably in the study design and population section. Please separate the methods from the results.

Response to comment #3:

We appreciate the reviewer's comment. We reorganized the methods and results section. Please see the revised version of our manuscript. We hope that our manuscript is now suitable for publication.

Comment #4: 

The methods states drugs of interest are included if they are prescribed more than 500 times and then later it says for >5% of study subjects. Can you please clarify? And please elaborate on how this approach has limitations as well. Although a drug may not be frequently prescribed, its risk of prolonging the QT interval persists and you excluded drugs infrequently prescribed.

Response to comment #4:

Both conditions were used to secure statistical power in each analysis. However, due to the different study designs, different conditions were adopted. In the case of a case-control study, usually, the case group is much smaller than the control group. If there is not enough drug prescription count in the case group, the 95% confidence interval is wide and becomes statistically non-significant even though the odds ratio is high. In retrospective cohort design, if there is not enough portion of exposure group, the comparison between exposure and non-exposure groups cannot be conducted. To clarify these points, we added the following text in the revised version.

• (Page 6) Further, we extracted all prescription data seven days before the date of ECG measurements in the QT prolongation cases, counting the frequency of each drug use. We also included all drugs to evaluate the QT prolongation risk of all drugs, including those not unknown yet. However, drugs prescribed more than 500 times in the cases were included to secure the statistical power.

• (Page 8) To secure at least 5% of the exposure group for further analysis, we selected 78 drugs prescribed for more than 5% of the study subjects as candidate drugs.

We also added the further possibility on the analysis on the drugs excluded in the study in the future as follows:

• (Page 16) Lastly, not all drugs used in the subject hospital were analyzed. It was to secure the least number of patients exposed to the target drug. If the drugs are more used, those drugs will be able to be included in our analysis in the future.

Comment #5: 

Please explain why the 18 drugs already classified by credmeds were excluded. Including them may have provided some triangulation to support your findings.

Response to comment #5:

In the retrospective cohort study design, it is critical to reduce potential bias by matching exposure and non-exposure group. To match each group as elaborately as possible, we included the prescription history of those 18 drugs in the matching variable. We added this description in the revised version as follows:

• (Page 8) Eighteen drugs were already classified as QT prolongation risk drugs by CredibleMeds; these are used for propensity score matching (see “Matching exposure and non-exposure groups”). Therefore, we finally analyzed the remaining 60 drugs.

Comment #6: 

Please provide a reference for your definitions of QTc prolongation. Clinical and research audiences often look to >/=500ms as a meaningful measure, but this does not necessarily align with all measures of abnormal QTc intervals.

Response to comment #6:

We appreciate the reviewer's comment. As per the comment, QTc>/=500 is also used. But the definition of QTc prolongation according to the American Heart Association (AHA), The American College of Cardiology Foundation (ACCF), Heart Rhythm Society (HRS), QTc longer than 450ms in men and 460 in women is recommended. QTc longer than 500 is usually used to define severe QTc prolongation. We added the following recommendations from AHA/ACCF/HRS and our previous studies used the same QTc definition in the reference list.

• (Page 6) To create the list of candidate drugs for analysis, we extracted all ECG results with QT prolongation (QTc>450 for men and QTc>460 for women) and randomly selected one ECG with several prolonged QT intervals for a single patient [17-19].

• (Page 8) We defined QT prolongation as QTc>450 in men and QTc>460 in women [17-19].

• (References) 17. Rautaharju PM, Surawicz B, Gettes LS, Bailey JJ, Childers R, Deal BJ, et al. AHA/ACCF/HRS recommendations for the standardization and interpretation of the electrocardiogram: part IV: the ST segment, T and U waves, and the QT interval: a scientific statement from the American Heart Association Electrocardiography and Arrhythmias Committee, Council on Clinical Cardiology; the American College of Cardiology Foundation; and the Heart Rhythm Society. Endorsed by the International Society for Computerized Electrocardiology. J Am Coll Cardiol. 2009;53(11):982-91. Epub 2009/03/14. doi: 10.1016/j.jacc.2008.12.014. PubMed PMID: 19281931.

• (References) 18. Choi BJ, Koo Y, Kim TY, Chung WY, Jung YJ, Park JE, et al. Risk of QT prolongation through drug interactions between hydroxychloroquine and concomitant drugs prescribed in real world practice. Sci Rep. 2021;11(1):6918. Epub 2021/03/27. doi: 10.1038/s41598-021-86321-z. PubMed PMID: 33767276; PubMed Central PMCID: PMCPMC7994840.

• (References) 19. Kim TY, Choi BJ, Koo Y, Lee S, Yoon D. Development of a Risk Score for QT Prolongation in the Intensive Care Unit Using Time-Series Electrocardiogram Data and Electronic Medical Records. Healthc Inform Res. 2021;27(3):182-8. Epub 2021/08/14. doi: 10.4258/hir.2021.27.3.182. PubMed PMID: 34384200; PubMed Central PMCID: PMCPMC8369048.

Comment #8: 

In the methods, please elaborate on the patient population from which 12-lead data was collected. Was is only ICU patients as was the case for continuous data?

Response to comment #8:

Thanks for this comment. We agree that the description of the study population for 12-lead data was not enough in the previous version. We modified the description to clarify the subject population as follows:

• (Page 5) Since we extracted data regardless of visit or admission type, all patients (outpatients, inpatients admitted to general wards/ICUs, patients visited emergence department, etc.) were included.

Comment #9: 

Please clarify whether the candidate drugs were evaluated before the ECG measurements (not 7 days after).

Response to comment #9:

We would like to thank the reviewer for this suggestion. As suggested, we modified the description to clarify the period included in our study as follows:

• (Page 6) Further, we extracted all prescription data seven days before the date of ECG measurements in the QT prolongation cases…

Comment #10: 

1st sentence of the 3rd paragraph of the introduction: It is unclear why “Standard 12-lead electrocardiogram (ECG) data” is explicitly called out. The subsequent limitations are not related to the 12-lead (versus continuous).

Response to comment #10:

We agree with this comment. That limitation is not limited to the 12-lead ECG data. Therefore, we modified the first part of the 3rd paragraph of the introduction as follows: 

• (Page 3) Many studies have examined drug-induced QT prolongation, but most of these investigations have the following limitations. In many cases, researchers first select the drug to be investigated [5, 8]. This approach enables researchers to investigate the risks for only some drugs of interest.

Comment #11: 

Please be sure you are using the acronym EMR correctly. An EMR and EHR are different and most health systems use an EHR, not EMR.

Response to comment #11:

EHR is correct. Thanks for correcting our mistake. We revised them as follows:

• (Page 5) The first comprised standard 12-lead ECG data [11]. We extracted the standard 12-lead ECG data from the ECG MUSE system (GE Healthcare) of Ajou University Hospital for the period between 1994 and 2018 and linked them to electronic health record (EHR) data.

• (Page 7) … we adopted the propensity score matching method to match the control group (subjects whose QTc interval was within normal range) with the case group (subjects whose QTc interval was prolonged) to adjust confounding variables with the following covariates: Gender and age at the ECG examination date, the latest serum potassium and calcium levels calculated within a year of the ECG measurement date, the comorbidities recorded in the EHR within a year of the ECG measurement date…

• (Page 8) For each covariate, all the drug infusions within 12 hours of the index time and all medical diagnosis records made before the index time were extracted from EHR data.

Comment #12: 

Please consider whether adherence or compliance is the more accurate term to describe how/if patients take medications.

Response to comment #12:

We would like to thank the reviewer for this suggestion. We noticed that ‘adherence’ has been used as a replacement for ‘compliance’, and ‘adherence’ is a more suitable term because it includes the meaning of patients’ autonomy.

• (Page 4) Since a patient’s drug adherence for the period between two measurements is unknown, it is difficult to capture the exact time gap between drug exposure and adverse drug events and identify acute adverse drug events.

---

## [Editor Report · Decision Letter 1]

13 Jan 2022

Data-driven drug-induced QT prolongation surveillance using adverse reaction signals derived from 12-lead and continuous electrocardiogram data

PONE-D-21-15865R1

Dear Dr. Yoon,

We’re pleased to inform you that your manuscript has been judged scientifically suitable for publication and will be formally accepted for publication once it meets all outstanding technical requirements.

Kind regards,

Chiara Lazzeri

Academic Editor

PLOS ONE
---

## [Editor Report · Acceptance letter]

21 Jan 2022

PONE-D-21-15865R1 

Data-driven drug-induced QT prolongation surveillance using adverse reaction signals derived from 12-lead and continuous electrocardiogram data 

Dear Dr. Yoon:

I'm pleased to inform you that your manuscript has been deemed suitable for publication in PLOS ONE. Congratulations! Your manuscript is now with our production department. 

Kind regards, 

on behalf of

Dr. Chiara Lazzeri 

Academic Editor

PLOS ONE